# Dynamical Wasserstein Barycenters for Time-series Modeling

**Kevin C. Cheng**
Tufts University
kevin.cheng@tufts.edu

**Shuchin Aeron**
Tufts University
shuchin.aeron@tufts.edu

**Michael C. Hughes**
Tufts University
michael.hughes@tufts.edu

**Eric L. Miller**
Tufts University
eric.miller@tufts.edu

## Abstract

Many time series can be modeled as a sequence of segments representing high-level discrete states, such as running and walking in a human activity application. Flexible models should describe the system state and observations in stationary "pure-state" periods as well as transition periods between adjacent segments, such as a gradual slowdown between running and walking. However, most prior work assumes instantaneous transitions between pure discrete states. We propose a dynamical Wasserstein barycentric (DWB) model that estimates the system state over time as well as the data-generating distributions of pure states in an unsupervised manner. Our model assumes each pure state generates data from a multivariate normal distribution, and characterizes transitions between states via displacement-interpolation specified by the Wasserstein barycenter. The system state is represented by a barycentric weight vector which evolves over time via a random walk on the simplex. Parameter learning leverages the natural Riemannian geometry of Gaussian distributions under the Wasserstein distance, which leads to improved convergence speeds. Experiments on several human activity datasets show that our proposed DWB model accurately learns the generating distribution of pure states while improving state estimation for transition periods compared to the commonly used linear interpolation mixture models.

## 1 Introduction

We consider the problem of estimating the dynamically evolving state of a system from time-series data.[1] The notion of "state" in such contexts typically is modeled in one of two ways. For many problems, the system state is a vector of continuous quantities (Kalman, 1960; Krishnan et al., 2016), perhaps constrained in some manner. Alternatively, discrete-state models take on one of a countable number of options at each point in time, as exemplified by hidden Markov models (HMMs) (Rabiner, 1989) or "switching-state" extensions (Ghahramani and Hinton, 2000; Linderman et al., 2017).

Many time-series characterization problems of current interest warrant a hybrid of continuous and discrete state representation approaches, where the system gradually transitions in a continuous manner among a finite collection of "pure" discrete states. For example, in human activity recognition using accelerometer sensor data (Bi et al., 2021), some segments of data do correspond to distinct activities (run, sit, walk, etc.), suggesting a discrete state representation. However, when using sensors with high-enough sampling rates, *transition* periods when the system is evolving from one state to another (e.g. the individual accelerates from standing to running over a few seconds) can be well

---

[1]Code available at https://github.com/kevin-c-cheng/DynamicalWassBarycenters_Gaussian

35th Conference on Neural Information Processing Systems (NeurIPS 2021).

resolved. Gradual evolution between pure states can also be observed in other domains of time series data, such as economics (Chang et al., 2016) or climate science (Chang et al., 2020). Characterizing these systems requires a model with a continuous state space to capture the gradual evolution of the system among the discrete set of pure states.

Motivated by this class of applications, we consider models for time series in which the system's dynamical state is specified by a vector of convex combination weights for mixing a set of data-generating distributions that define the individual pure states. Many approaches, such as mixture models, interpret such a simplex-constrained state vector (Rudin, 1976) as assignment probabilities; that is, the system is assumed to be in a pure state with uncertainty as to which. As a result, the data-generating distribution at moments of transition is a convex, *linear* combination of the pure-state emission distributions. While useful in many applications, such linear interpolation does not capture the gradual transitions among pure states in the time series of interest to us.

To illustrate the shortcomings of linear interpolation, consider the toy data task in Fig. 1, where a system gradually transitions between three pure states over time. During the transition periods (e.g. at times 600 and 1400), the linear interpolation method infers a data-generating distribution that is *multi-modal*, shown in Fig. 1(b). If we refer to our pure states as "walk" and "run," this approach models the walk-to-run transition as sometimes walk and sometimes run. This does not intuitively capture the gradual nature of accelerating from walk to run in our intended applications.

To overcome this limited representation, we consider another way to mix together pure-state distributions: *displacement-interpolation* (McCann, 1997), which is related to the Wasserstein distance (Peyré and Cuturi, 2019), a metric over the space of probability distributions (Sriperumbudur et al., 2010). While the work of McCann (1997) is limited to combining two distributions, it is extended to multiple distributions using the notion of a *Wasserstein barycenter* (Agueh and Carlier, 2011). Fig. 1(c) shows how a Wasserstein barycenter approach to time-series modeling infers data-generating distributions during transitions that are not multi-modal but instead place mass *in between* where the two pure-state distributions do. This intuitively captures *gradual transition* between two pure states.

Inspired by this framework, in this work we develop a dynamical Wasserstein barycentric (DWB) model for time series intended to explain data arising as a system evolves between pure states. Our model uses a barycentric weight vector to represent the system state. Given an observed multivariate time-series and a desired number of states $K$, all parameters are estimated in an unsupervised way. Estimation simultaneously learns the data-generating distributions of $K$ pure discrete states as well as the $K$-simplex valued barycentric weight vector state at each timestep.

Given the nature of our model, we require that the state lie in the simplex at every timestep, a constraint not respected by the Gaussian noise that drives common continuous-state processes (Welch, 1997). Building on work by Nguyen and Volkov (2020), we employ a random walk where the driving noise comes from independent, identically distributed (IID) draws from a mixture of two Beta distributions, representing stationary and transitional dynamics. By blending the current state and a mixture-of-Betas draw in a convex manner, we construct a new state that lies in the simplex.

To specify the emission distributions of our model, we assume that each pure state generates data from a multivariate Gaussian. While a Gaussian model may not be suitable in all applications, this choice allows us to exploit useful properties of Gaussian densities under the Wasserstein distance (Takatsu, 2011). Specifically, a closed-form expression exists for the Wasserstein distance between Gaussians, the Wasserstein barycenter among Gaussians can be computed via a simple recursion, and the estimation of the Gaussian mean vectors and covariance matrices can be performed conveniently over a Riemannian product manifold. Empirically, we find our proposed DWB model with Gaussian pure-states performs well on human activity datasets, accurately characterizing both pure-states emission distributions and capturing the system state in pure states and transition periods.

**Contributions:** We introduce a *displacement-interpolation model for time series* where the data-generating distribution is given by the weighted Wasserstein barycenter of a set of pure-state emission distributions and a time-varying state vector. We propose a *simplex-valued random walk* with flexible dynamical structure to model the system state. We exploit the *Riemannian structure of Gaussian distributions* under the Wasserstein distance for parameter estimation for faster convergence speed. We evaluate on *human activity data* and demonstrate the ability of our method to capture *stationary and transition dynamics*, comparing with the linear interpolation mixture model and with a continuous state space model.

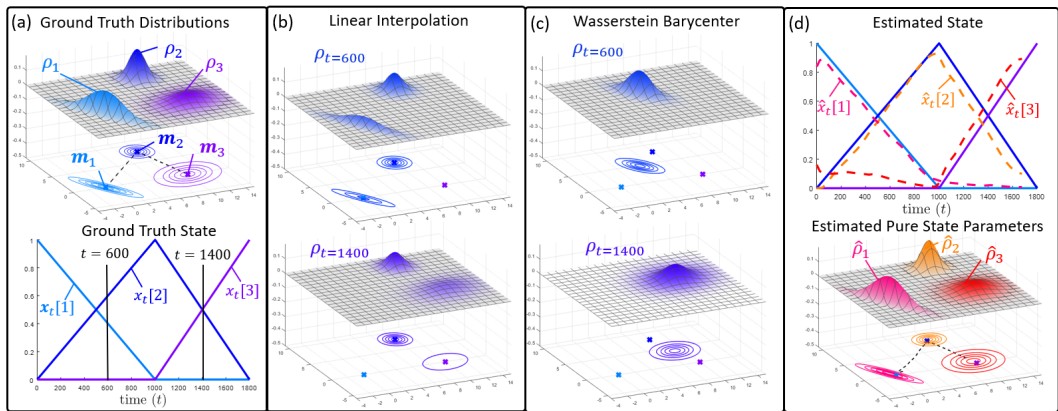

Figure 1: (a) Three Gaussian distributions $\rho_1, \rho_2, \rho_3$ each representing distinct activities with corresponding means $m_1, m_2, m_3$ that are marked ('x') in all plots as reference points. The time series is drawn from a time-varying distribution according to the ground truth state vector as the system transitions linearly from $\rho_1$ to $\rho_2, t = 1, ..., 1000$, then continues to $\rho_3, t = 1000, ..., 1800$. (b) Under the linear interpolation state-transition model, the PDF at select times of $t = 600, 1400$ are linear combinations of $\rho_1, \rho_2, \rho_3$. (c) Alternatively, the proposed displacement-interpolation transition model between pure-states given by the Wasserstein barycenter translates the mass between pure states. (d) Following the Wasserstein barycentric model for time series, our proposed method accurately recovers both the pure state distributions and state vector from the observed time series.

**Outline**: Sec. 2 provides an overview of the Wasserstein distance, barycenter, and the associated geometry for Gaussian distributions. We then formalize our problem statement and estimation problem in Sec. 3. Sec. 4 discusses the model parameters, covering the dynamical simplex state-space model in Sec. 4.1 and the pure-state parameters in Sec. 4.2. Sec. 5 discusses the optimization of our model parameters leveraging geometric properties of the Wasserstein distance for Gaussians. Finally, Sec. 6 evaluates and discusses the advantages of our model in the context of human activity data.

## 2 Technical Background

A core component to our approach is to model the intermediate transition states of a time series using the Wasserstein barycenter (Agueh and Carlier, 2011) of probability distributions, which generalizes the displacement-interpolation framework (McCann, 1997) beyond two distributions. We refer the works of (Peyré and Cuturi, 2019) and (Villani, 2009) for a detailed discussion on these concepts.

Consider the space of all Borel probability measures over $\mathbb{R}^d$ with finite second moment. The squared Wasserstein-2 distance for two distributions $\rho_1, \rho_2$ with squared Euclidean ground cost is defined as,

$$\mathcal{W}_2^2(\rho_1, \rho_2) = \inf_{M \in \Pi(\rho_1, \rho_2)} \int_{\mathbb{R}^d \times \mathbb{R}^d} \|\boldsymbol{\alpha} - \boldsymbol{\beta}\|_2^2 \, M(d\boldsymbol{\alpha}, d\boldsymbol{\beta}), \tag{1}$$

where $\Pi(\rho_1, \rho_2)$ is the set of all joint distributions with marginals $\rho_1, \rho_2$, and $M$ is the optimal transport plan, the element that minimizes the total transportation cost. When these measures are Gaussian, parameterized by their mean vectors $m_i \in \mathbb{R}^d$, and symmetric positive-definite covariance matrices $S_i \in Sym_+^d$, the squared Wasserstein-2 distance has a closed form solution (Takatsu, 2011),

$$\mathcal{W}_2^2\left(\rho_1(m_1, S_1), \rho_2(m_2, S_2)\right) = \underbrace{\|m_1 - m_2\|_2^2}_{\mathcal{E}^2(m_1, m_2)} + \underbrace{\mathrm{tr}\left(S_1 + S_2 - 2\left(S_1^{\frac{1}{2}} S_2 S_1^{\frac{1}{2}}\right)\right)}_{\mathcal{B}^2(S_1, S_2)}. \tag{2}$$

This distance decomposes into sum of the squared Euclidean distance between mean vectors, $\mathcal{E}^2(m_1, m_2)$, and the squared Bures distance (Bhatia et al., 2017) between covariance matrices, $\mathcal{B}^2(S_1, S_2)$. Thus, the contributions of the mean and covariance to the Wasserstein distance between Gaussians are decoupled, a property that is uncommon for Gaussian distribution distances (Nagino and Shozakai, 2006) and has important implications for optimization and barycenter computation.

The Wasserstein barycenter extends the notion of the weighted average of points in $\mathbb{R}^d$ using the Euclidean distance to the space of probability distributions with the Wasserstein distance. Given a set of $K$ measures and the barycentric coordinate vector on the $K$-simplex, $\boldsymbol{x} \in \Delta^K$, the Wasserstein

barycenter is the measure that minimizes this weighted Wasserstein distance to the set of measures,

$$\rho_B\left(\boldsymbol{x}, \{\rho_k\}_{k=1}^K\right) = \operatorname*{argmin}_{\rho \in \mathcal{P}_2(\mathbb{R}^d)} \sum_{k=1}^K \boldsymbol{x}[k]\mathcal{W}_2^2(\rho_k, \rho). \quad (3)$$

When $\rho_k$ are Gaussian distributions, $\rho_B$ defined in (3) is itself Gaussian (Agueh and Carlier, 2011) with parameters $\boldsymbol{m}_B, \boldsymbol{S}_B$. Again, because of the decomposition of the Wasserstein distance in (2), the Wasserstein barycentric problem in (3) can be solved separately for its components,

$$\boldsymbol{m}_B = \operatorname*{argmin}_{\boldsymbol{m} \in \mathbb{R}^d} \sum_{k=1}^K \boldsymbol{x}[k]\mathcal{E}^2(\boldsymbol{m}_k, \boldsymbol{m}), \qquad \boldsymbol{S}_B = \operatorname*{argmin}_{\boldsymbol{S} \in Sym_+^d} \sum_{k=1}^K \boldsymbol{x}[k]\mathcal{B}^2(\boldsymbol{S}_k, \boldsymbol{S}). \quad (4)$$

The optimal mean can be computed in closed-form: $\boldsymbol{m}_B = \sum_k \boldsymbol{x}[k]\boldsymbol{m}_k$. The optimal covariance matrix can be solved via the fixed-point iteration proposed in Álvarez Esteban et al. (2016).

## 3  Problem Formulation

**Model Definition.** Our DWB model's data-generating process is illustrated in Fig. 2. First, the *pure-state emission parameters* $\boldsymbol{\Theta} \equiv \{(\boldsymbol{m}_k, \boldsymbol{S}_k)\}_{k=1}^K$, define a Gaussian distribution $\rho_k$ for each pure state $k$. Second, the *state vector* $\boldsymbol{x}_t$ defines the system state at $t$ and lies on the simplex. Given $\boldsymbol{\Theta}$ and $\boldsymbol{x}_t$, we can form a time-varying Gaussian distribution $\rho_{B_t}(\boldsymbol{x}_t, \boldsymbol{\Theta}) \equiv N(\boldsymbol{m}_{B_t}, \boldsymbol{S}_{B_t})$ for each $t$, which is a barycentric combination of the $K$ pure Gaussians using weights $\boldsymbol{x}_t$ via (4). We can write the states for an entire sequence as $\boldsymbol{X}$, comprised of an initial state and a sequence of simplex-valued state vectors, denoted $\boldsymbol{X} \equiv \{\boldsymbol{x}_0, \{\boldsymbol{x}_t\}_{t=1}^T\}$.

**Data Preprocessing.** We are given a vector-valued time series of observations $\boldsymbol{y}_\tau \in \mathbb{R}^d, \tau = 1, ..., \mathcal{T}$. Instead of modeling this data directly, to improve smoothness we model the empirical distribution of sliding windows of $2n+1$ samples (Aghabozorgi et al., 2015) strided by $\delta$ samples. We retain only windows with complete data, with

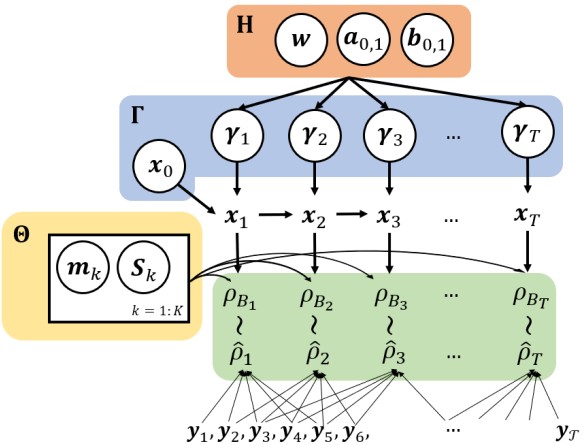

Figure 2: Graphical model diagram of our proposed dynamic Wasserstein barycenter (DWB) time series model. For each window of data, indexed by $t$, our model forms an emission distribution $\rho_{B_t}$ that is the Wasserstein barycenter given known pure-state Gaussian parameters $\boldsymbol{\Theta} = \{\boldsymbol{m}_k, \boldsymbol{S}_k\}_{k=1}^K$ and time-varying weight vector $\boldsymbol{x}_t$. The simplex-valued state sequence $\{\boldsymbol{x}_t\}_{t=1}^T$ is drawn from a random walk using Beta-mixture draws $\boldsymbol{\gamma}_t$ (Sec. 4.1), with hyperparameters $\boldsymbol{H} = \{\boldsymbol{w}, \boldsymbol{a}_{0,1}, \boldsymbol{b}_{0,1}\}$ and initial state $\boldsymbol{x}_0$. Random variables are denoted by circles. Figure shown has $n = 2, \delta = 1$ Colors correspond to terms in (9).

start times corresponding to $\tau = 1, (\delta + 1), (2\delta + 1), ..., \lfloor \frac{(\mathcal{T} - (2n+1))}{\delta} \rfloor\delta + 1$, which we index sequentially as $t \in \{1, 2, \ldots T\}$. A window indexed at $t$ corresponds to a window centered at $\tau = (\delta(t-1) + n + 1)$, which provides an estimates of the underlying distribution at $\boldsymbol{y}_\tau$. For each window location $t$, we compute an unbiased *empirical* Gaussian distribution $\hat{\rho}_t = \mathcal{N}(\boldsymbol{m}_t, \boldsymbol{S}_t)$ where,

$$\boldsymbol{m}_t = \frac{1}{2n+1} \sum_{i=1}^{2n+1} \boldsymbol{y}_{\delta(t-1)+i}, \qquad \boldsymbol{S}_t = \frac{1}{2n} \sum_{i=1}^{2n+1} (\boldsymbol{y}_{\delta(t-1)+i} - \boldsymbol{m}_t)(\boldsymbol{y}_{\delta(t-1)+i} - \boldsymbol{m}_t)^T. \quad (5)$$

Minimizing the Wasserstein distance between this sequence of empirical distributions $\hat{\rho}_t$ and the model-predicted distributions $\rho_{B_t}$ drives our model's parameter learning.

**Estimation Objective.** In practice, we are given an observed sequence of empirical distributions $\{\hat{\rho}_t\}_{t=1}^T$ and a desired number of states $K$. We wish to estimate the state sequence $\boldsymbol{X}$ and emission parameters $\boldsymbol{\Theta}$. We pose the estimation of $\boldsymbol{X}$ and $\boldsymbol{\Theta}$ as the solution to an optimization problem seeking to balance fidelity to a prior model on the parameters of interest with the desire to minimize the time integrated Wasserstein distance between the predicted and observed distributions:

$$\hat{\boldsymbol{X}}, \hat{\boldsymbol{\Theta}} = \operatorname*{argmin}_{\boldsymbol{X}, \boldsymbol{\Theta}} -\log\left(p(\boldsymbol{X})p(\boldsymbol{\Theta})\right) + \lambda \sum_{t=1}^T \mathcal{W}_2^2(\hat{\rho}_t, \rho_{B_t}(\boldsymbol{x}_t, \boldsymbol{\Theta})). \quad (6)$$

The scalar weight $\lambda > 0$ trades off the model's fit to data (measured by the Wasserstein distance) with the probability of the state sequence $\boldsymbol{X}$ and pure-state emission parameters $\boldsymbol{\Theta}$ under assumed *prior* distributions. Our chosen priors $p(\boldsymbol{X})$ and the $p(\boldsymbol{\Theta})$ are covered in the following section.

# 4 Model Parameter Priors

## 4.1 Prior on Simplex States over Time

Here we develop the *transition* model that generates the sequence of state vectors $\boldsymbol{x}_0, \boldsymbol{x}_1, \ldots \boldsymbol{x}_T$. We assume a first-order Markovian structure: $p(\boldsymbol{X}) = p(\boldsymbol{x}_0) \prod_{t=1}^{T} p(\boldsymbol{x}_t | \boldsymbol{x}_{t-1})$. Recall that each state vector lies on the $K$-dimensional *simplex*. The geometry of the state space in the case of $K = 3$ states is shown in Fig. 3, where $\boldsymbol{x}_t$ lies in the convex hull of the three simplex vertices, the unit coordinate vectors $\boldsymbol{e}_1, \boldsymbol{e}_2, \boldsymbol{e}_3$. Each vertex is associated with a pure-state in our problem. For a more general $K$-state problem, this is generalized to the $K$−dimensional simplex, denoted $\Delta^K$, in a straightforward manner.

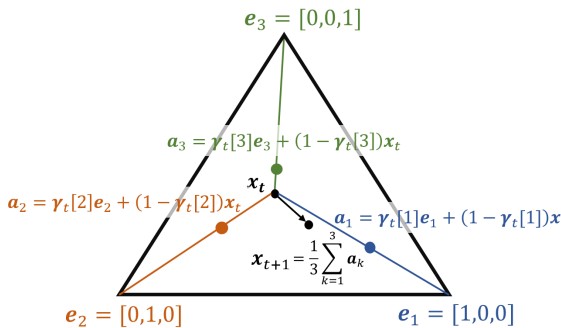

Figure 3: Given current state $\boldsymbol{x}_t$, we transition to next state $\boldsymbol{x}_{t+1}$ by averaging $K$ step-to-vertex updates. For each $k = 1, ..., K$, step-length $\boldsymbol{\gamma}_t[k] \in [0, 1]$ represents a proportional step from $\boldsymbol{x}_t$ to simplex vertex $\boldsymbol{e}_k$.

To ensure that each new state $\boldsymbol{x}_t$ lies in the simplex, we define its update using a $K$-dimensional "innovations" vector, $\boldsymbol{\gamma}_t$. As seen in Fig. 3, we imagine taking $K$ different steps from the previous state $\boldsymbol{x}_{t-1}$. Each step (indexed by $k$) moves toward vertex $\boldsymbol{e}_k$ with proportional step length $\boldsymbol{\gamma}_t[k] \in [0, 1]$. A zero-length step ($\boldsymbol{\gamma}_t[k] = 0$) leaves the state at its previous value $\boldsymbol{x}_{t-1}$ while a full step ($\boldsymbol{\gamma}_t[k] = 1$) jumps to the vertex $\boldsymbol{e}_k$. Unlike prior methods (Nguyen and Volkov, 2020), we repeat this process for each of the $K$ components and average their results to achieve the next state $\boldsymbol{x}_t$,

$$\boldsymbol{x}_t = (1 - \tfrac{1}{K} \sum_{k=1}^{K} \boldsymbol{\gamma}_t[k]) \boldsymbol{x}_{t-1} + \tfrac{1}{K} \boldsymbol{\gamma}_t. \tag{7}$$

By construction, (7) delivers a valid state $\boldsymbol{x}_t$ that lies in the $K$-simplex.

Inspired by ideas from dynamical Bayesian nonparametric models (Ren et al., 2008), a suitable prior over innovations $\boldsymbol{\gamma}_t$ on the domain $[0, 1]$ is the Beta distribution (Yates and Goodman, 2005). We draw independent innovation values for each component (indexed by K) as IID across time according to a two-component Beta *mixture*. The first component (index 0) captures *stationary* behavior and the second component (index 1) captures *transitions* between pure states:

$$p\left(\{\boldsymbol{\gamma}_t\}_{t=1}^T\right) = \prod_{t=1}^{T} \prod_{k=1}^{K} \boldsymbol{w}[k] \text{Beta}\left(\boldsymbol{\gamma}_t[k]; \boldsymbol{a_0}[k], \boldsymbol{b_0}[k]\right) + (1 - \boldsymbol{w}[k]) \, \text{Beta}\left(\boldsymbol{\gamma}_t[k]; \boldsymbol{a_1}[k], \boldsymbol{b_1}[k]\right). \tag{8}$$

This Beta-mixture prior for $\boldsymbol{\gamma}_t$ allows flexibility in how $\boldsymbol{x}_t$ evolves on the simplex. By requiring that the Beta parameters of each component are larger than one ($\boldsymbol{a}[k] > 1, \boldsymbol{b}[k] > 1$), we induce uni-modal distributions on $[0, 1]$. For the stationary component (index 0), we expect innovations close to zero, which means we should set $\boldsymbol{b}_0[k] \gg \boldsymbol{a}_0[k]$. We fix $\boldsymbol{a}_0[k] = 1.1, \boldsymbol{b}_0[k] = 20$ in all experiments. For the transition component (index 1) of each state $k$, we allow Beta parameters $\boldsymbol{a}_1[k], \boldsymbol{b}_1[k]$ as well as the mixture weight $\boldsymbol{w}[k]$ to be *learnable* hyperparameters, denoted $\boldsymbol{H} = \{\boldsymbol{w}, \boldsymbol{a}_1, \boldsymbol{b}_1\}$. To prevent mode collapse we constrain $\boldsymbol{a}_1[k] > 1.1$, $\frac{\boldsymbol{a}_1[k]}{\boldsymbol{a}_1[k] + \boldsymbol{b}_1[k]} > 0.15$, and $\boldsymbol{w}[k] \in [0.01, 0.99]$.

As mentioned in Sec. 3, the simplex-state $\boldsymbol{x}_t$ represents the barycentric mixing weights used to compute the model-predicted distribution of the data. In our current formulation, this sequence of states is *deterministic* according to (7), given the initial state and the sequence of innovation vectors. Since these innovations are the random variables of interest, it is convenient to replace $\boldsymbol{X}$ in (6) with $\boldsymbol{\Gamma} \equiv \left\{ \boldsymbol{x}_0, \{\boldsymbol{\gamma}_t\}_{t=1}^T \right\}$ resulting in the estimation problem,

$$\hat{\boldsymbol{\Gamma}}, \hat{\boldsymbol{\Theta}}, \hat{\boldsymbol{H}} = \underset{\boldsymbol{\Gamma}, \boldsymbol{\Theta}, \boldsymbol{H}}{\operatorname{argmin}} \underbrace{-\log\left( p_{\boldsymbol{H}}(\boldsymbol{\Gamma}) \; p(\boldsymbol{\Theta}) \right) + \lambda \sum_{t=1}^{T} \mathcal{W}_2^2(\hat{\rho}_t, \rho_{B_t}(\boldsymbol{\Gamma}, \boldsymbol{\Theta}))}_{F\left(\boldsymbol{\Gamma}, \boldsymbol{\Theta}, \boldsymbol{H}, \{\hat{\rho}_t\}_{t=1}^T\right)}. \tag{9}$$

In our implementation, we are indifferent to the initial state and therefore set $p(\boldsymbol{x}_0)$ as uniform over the simplex. The coloration in (9) is linked to the associated parameters in Fig. 2.

## 4.2 Prior on Pure-State Emission Parameters

The final component to (9) is $p(\boldsymbol{\Theta})$, the prior on the pure-state emission parameters. Using a reference normal distribution $\mathcal{N}(\boldsymbol{m}_0, \sigma_0 \boldsymbol{I})$, we can define a probability density function over the space of all Gaussian distributions derived from the Wasserstein distance to this reference distribution,

$$p(\boldsymbol{m}, \boldsymbol{S}) = \kappa(s, \sigma_0) \exp\left(-\frac{1}{2s^2} \mathcal{W}_2^2\left((\boldsymbol{m}, \boldsymbol{S}), (\boldsymbol{m}_0, \sigma_0^2 \boldsymbol{I})\right)\right) \tag{10}$$

$$= \kappa(s, \sigma_0) \exp\left(-\frac{1}{2s^2} \|\boldsymbol{q} - \boldsymbol{q}_0\|_2^2\right),$$

where $s$ is a scalar hyperparameter that controls the variance of this prior. A simple calculation shows that (10) is equivalent to a multivariate Gaussian distribution over $\boldsymbol{q} \equiv [\boldsymbol{m}, \boldsymbol{\omega}] \in \mathbb{R}^{2d}$, the joint space of means $\boldsymbol{m}$ and the eigenvalues $\boldsymbol{\omega}$ of the covariance matrices $\boldsymbol{S} \in Sym_+^d$. This Gaussian has mean $\boldsymbol{q}_0 = [\boldsymbol{m}_0, \sigma_0, ..., \sigma_0]$ and covariance equal to $s\boldsymbol{I}$. Since the eigenvalues of $\boldsymbol{S}$ must be positive, it follows that the normalizing constant needed for (10) to be a valid distribution given the normal CDF function $\Phi$ is $\kappa(s, \sigma_0) = \left((2\pi s^2) \Phi\left(\frac{\sigma_0}{s}\right)\right)^{-d}$. We assume that the pure-state distributions parameters are mutually independent. To ensure that this prior scales similarly to the other terms in (9), all of which scale with the length of the time series $T$, we set $\log(p(\boldsymbol{\Theta})) = T \sum_{k=1}^{K} \log(p(\boldsymbol{m}_k, \boldsymbol{S}_k))$.

# 5 Model Estimation

---

**Algorithm 1:** Dynamical Wasserstein Barycenter (DWB) Time-Series Estimation

---

**Input:**
$\boldsymbol{y}_\tau, \tau = 1 \ldots \mathcal{T}$: Time series observations
$K$: Number of pure states

**Hyperparameters:**
$n$: Window size,    $\delta$: Window stride
$\lambda$: Weight on data-fit term
$s$: Variance on prior for $\boldsymbol{\Theta}$
$(\mu_0, \sigma_0)$: Mean, var. of $p(\boldsymbol{\Theta})$ reference dist.
$\eta$: Convergence threshold

**Output:**
$\boldsymbol{\Theta} = \left\{\{\boldsymbol{m}_k, \boldsymbol{S}_k\}_{k=1}^{K}\right\}$: Pure-state emission params
$\boldsymbol{\Gamma} = \left\{\boldsymbol{x}_0, \{\boldsymbol{\gamma}_t\}_{t=1}^{T}\right\}$: Initial state and innovations
$\boldsymbol{X} = \{\boldsymbol{x}_t\}_{t=1}^{T}$: Wasserstein barycentric state vector
                            (Computed from $\boldsymbol{\Gamma}$ via (7))
$\boldsymbol{H} = \{\boldsymbol{w}, \boldsymbol{a}_1, \boldsymbol{b}_1\}$: Beta mixture parameters for
                            transition dynamics

---

1 **for** $t = 1, ..., T$ where $T = \lfloor \frac{(\mathcal{T} - (2n+1))}{\delta} \rfloor + 1$ **do**
2 $\quad \boldsymbol{m}_t = \frac{1}{(2n+1)} \sum_{i=1}^{2n+1} \boldsymbol{y}_{\delta(t-1)+i}$ ;                    // Preprocessing of windowed
3 $\quad \boldsymbol{S}_t = \frac{1}{2n} \sum_{i=1}^{2n+1} (\boldsymbol{y}_{\delta(t-1)+i} - \boldsymbol{m}_t)(\boldsymbol{y}_{\delta(t-1)+i} - \boldsymbol{m}_t)^T$ ;   // empirical distributions
4 $\quad \hat{\rho}_t = \mathcal{N}(\boldsymbol{m}_t, \boldsymbol{S}_t)$
5 **end**
6 $c^{(0)} = F\left(\boldsymbol{\Gamma}^{(0)}, \boldsymbol{\Theta}^{(0)}, \boldsymbol{H}^{(0)}, \{\hat{\rho}_t\}_{t=1}^{T}\right)$ ;                    // Cost function $F$ defined in (9)
7 **do**
8 $\quad \boldsymbol{\Gamma}^{(n+1)}, \boldsymbol{H}^{(n+1)} = \operatorname{argmin}_{\boldsymbol{\Gamma}, \boldsymbol{H}} F\left(\boldsymbol{\Gamma}^{(n)}, \boldsymbol{\Theta}^{(n)}, \boldsymbol{H}^{(n)}, \{\hat{\rho}_t\}_{t=1}^{T}\right)$ ;                    // Adam
9 $\quad \boldsymbol{\Theta}^{(n+1)} = \operatorname{argmin}_{\boldsymbol{\Theta}} F\left(\boldsymbol{\Gamma}^{(n+1)}, \boldsymbol{\Theta}^{(n)}, \boldsymbol{H}^{(n+1)}, \{\hat{\rho}_t\}_{t=1}^{T}\right)$ ;   // Riemannian line search
10 $\quad c^{(n+1)} = F\left(\boldsymbol{\Gamma}^{(n+1)}, \boldsymbol{\Theta}^{(n+1)}, \boldsymbol{H}^{(n+1)}, \{\hat{\rho}_t\}_{t=1}^{T}\right)$
11 **while** $(c^{(n)} - c^{(n+1)}) > \eta$;

---

Given a desired number of states $K$ and a multivariate time series dataset $\boldsymbol{y}$, Alg. 1 details the steps needed to learn all parameters of our DWB model: $\boldsymbol{\Gamma}$, the initial state and innovations sequence that drive the dynamical state model; $\boldsymbol{\Theta}$, the pure-state emission distribution means and covariance matrices; and $\boldsymbol{H}$, the hyperparameters governing transition dynamics on the simplex.

The algorithm performs coordinate descent (updating some variables while fixing others) to optimize the objective function in (9). We chose this structure because the update to $\boldsymbol{\Theta}$ is able to exploit specialized optimization structure. Gradient descent methods are used to implement each minimization step in Alg. 1 taking advantage of auto-differentiation in PyTorch (Paszke et al., 2017). The runtime cost of each step in Alg. 1 is $\mathcal{O}(TKd^3)$, where $d$ is the dimension of each observed data vector $\boldsymbol{y}_t$.

**Updates to $\mathbf{\Gamma}, \mathbf{H}$ via Adam.** The Adam optimizer (Kingma and Ba, 2017) is used to solve the $\mathbf{\Gamma}, \mathbf{H}$ problem on line 8 of Alg. 1. To ensure that $\gamma_t \in [0, 1]$ for $t = 1, \ldots, T$. we clamp these parameters to $[\epsilon, 1 - \epsilon]$ for $\epsilon = 1e^{-6}$. The initial state vector is clamped and normalized to stay on the simplex in a similar manner and the parameters of $\mathbf{H}$ are clamped as mentioned in Sec. 4.1.

**Updates to $\mathbf{\Theta}$ via natural Riemannian geometry.** The pure-state emission parameters $\mathbf{\Theta}$ define the mean and covariance parameters for $K$ Gaussian distributions. While a variety of methods each based on different geometries have been proposed for optimizing Gaussian parameters (Lin, 2019; Hosseini and Sra, 2015; Arsigny et al., 2007), in this work we choose to leverage the geometry of Gaussian distributions under the Wasserstein distance (Malagò et al., 2018). From the decomposition in (2), we see that optimization for $\mathbf{\Theta} \equiv \{(\boldsymbol{m}_k, \boldsymbol{S}_k)\}_{k=1}^K$ under Wasserstein geometry can be carried out over a Riemannian product manifold $\left(\mathbb{R}^d \times Sym_+^d\right)$ (Hu et al., 2020) with standard Euclidean geometry on $\mathbb{R}^d$, and Wasserstein-Bures geometry on $Sym_+^d$ (Malagò et al., 2018; Takatsu, 2011; Bhatia et al., 2017). Therefore, we estimate $\mathbf{\Theta}$ over this Riemannian product manifold using a gradient descent line search algorithm (Absil et al., 2008). The supplement provides further details and experimental results demonstrating improved optimization speeds compared to standard Euclidean geometry.

# 6   Real World Results

## 6.1   Datasets and Evaluation Procedures

**Datasets.** Our work is motivated by applications in human activity accelerometry where "pure" states correspond to atomic actions such as walking, running, or jumping. We evaluate our algorithm on two datasets where smooth transitions between states are observable and the number of states is known.

*Beep Test (BT, proprietary):*  46 subjects run between two points to a metronome with increasing frequency. In this setting the subject alternates between running and standing thus we estimate a two state model. Data is captured from a three-axis accelerometer sampled at 100 Hz.

*Microsoft Research Human Activity (MSR, Morris et al. (2014)):* 126 subjects perform exercises in a gym setting. Exercises vary among subjects covering strength, cardio, cross-fit, and static exercises. Each time series is truncated to five minutes. We set $K$ to the number of labeled discrete states in the truncated time series (range: 2 to 7). The three-axis accelerometer is sampled at 50 Hz.

|  | BT | MSR |
|---|---|---|
| $n$ | 100 | 250 |
| $\delta$ | 25 | 125 |
| $\lambda$ | 100 | 100 |
| $s$ | 1.0 | 1.0 |
| $\eta$ | 1e-4 | 1e-4 |

Table 1: Model hyperparameters

**Available labels.** All models are trained in *unsupervised* fashion: each method is provided only the 3-axis accelerometer signal $\boldsymbol{y}$ and desired number of states $K$ as input. While some ground-truth state annotations are available, each timestep is labeled as belonging exclusively to one discrete state. This assumes *instantaneous* transition between pure states and belies the underlying gradual transitions (e.g. acceleration from stand to run) that actually occur in the data stream, which our method is designed for. Because annotations that properly characterize the gradual transition between states are not available, we evaluate performance based on how well a given model's predicted emission distribution fits the observed data over the whole time series.

**Performance metrics:** We measure data fit quality using both the average Wasserstein error (11), akin to the model-fit term in (9), as well as the negative log likelihood (12) of all samples in each window given the model's inferred barycentric distribution for that window.

$$e_W = \frac{1}{T} \sum_{t=1}^T \mathcal{W}_2^2(\hat{\rho}_t, \rho_{B_t}), \quad (11) \qquad e_{nll} = \frac{1}{T(2n+1)} \sum_{t=1}^T \sum_{i=1}^{2n+1} -\log\left(\rho_{B_t}(\boldsymbol{y}_{\delta(t-1)+i})\right). \quad (12)$$

**Baseline methods:** To the best of our knowledge, the problem of characterizing continuous transitions among discrete pure states is largely unexplored. Most relevant are continuous state space models which identify a continuous latent state, but not in a manner that identifies pure-states of the system. Therefore, we compare our proposed DWB model a continuous-state deep neural state space (DSS) model. Additionally, we baseline the Wasserstein barycentric interpolation model against the linear interpolation model given by discrete-state Gaussian mixture models (GMM).

*GMM Linear interpolation baseline.* Under the linear interpolation model, each timestep's emission distribution is a Gaussian mixture of the pure states, $\rho_{G_t} = \sum_{k=1}^K \boldsymbol{x}_t[k]\mathcal{N}(\boldsymbol{m}_k, \boldsymbol{S}_k)$. We highlight that $\rho_{B_t}$ and $\rho_{G_t}$ are equivalent when the $\boldsymbol{x}_t$ is in a pure-state, thus the difference between models

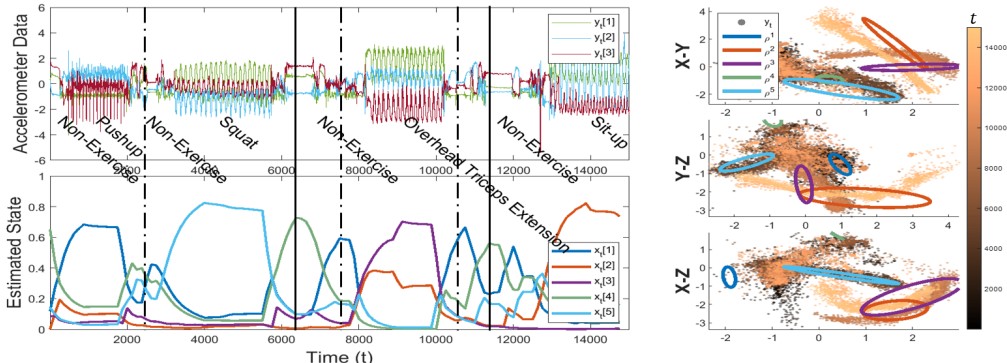

Figure 4: (top left) Three-axis accelerometer of a sample MSR time series consisting of 5 actions. (right) Estimated pure-state Gaussian distributions projected onto each pairwise axes. (bot left) The Wasserstein barycentric weights are given by the estimated system state.

lies in the *transition regions*. The GMM model is implemented in our framework by replacing $\rho_{B_t}$ with $\rho_{G_t}$ in Eqs. (9), (11), and (12). To compute the Wasserstein distance between a Gaussian and Gaussian mixture, we use the upper bound in Chen et al. (2018) for fast gradient-based parameter estimation, but use Monte-Carlo estimation (Sriperumbudur et al., 2010) for more precise evaluation.

*Deep neural State Space (DSS, (Krishnan et al., 2016)).* The DSS method uses neural networks to parameterize the transition dynamics and the emission distribution of the latent state. Unlike our DWB approach, continuous state space models do not identify pure-states of the system. Thus, post-processing of the latent state space would be required to identify such pure states under these frameworks. For our DWB model, the maximum number of parameters required for state transitions and emissions for the MSR dataset is $p = 91$ ($\mathcal{O}(d^2 K)$). Therefore, we evaluate the DSS model under two different settings, one where *both* the transition and emission networks are given a comparable number of parameters to the DWB model ($p = 94$) and one with many more ($p \approx 88,000$) parameters. Exact configuration details for DSS are provided in the supplement.

**Hyperparameters and initialization.** Hyperparameter choices are documented in Tab. 1, with window size $n$ chosen to capture 2-5 seconds of real-time activity in each dataset. We set $s = 1.0$ and $\lambda = 100$ according to the parameter selection study later in Sec. 6.2. We set $\boldsymbol{m}_0$ to the mean of the observed data, and $\sigma_0$ to the average eigenvalue of the set of covariance matrices obtained from a $K$-component GMM fit to the data using expectation-maximization via code from Pedregosa et al. (2011). Our $\boldsymbol{\Theta}$ estimation problem is non-convex, so the solution is dependent on initialization. To ensure fair comparison, we use the same initialization for each method: a time-series clustering method (Cheng et al., 2020a) that applies matched-filtered change point detection (Cheng et al., 2020b) for the Wasserstein distance. Additional details regarding optimization and initialization are provided in the supplement.

## 6.2 Experimental Results and Analysis

**Qualitative evaluation.** Fig. 4 shows for one exemplary MSR time series how our model can estimate pure-state emission distributions for the five states (*right*) and capture the system state in both the stationary and transition periods over time (*bot left*). Of interest are the segments of "Non-Exercise" that have significant contributions from "Pushup" (e.g. *dashed* lines at $t = 2.5k, 7.5k, 10.5k$). The data shows these regions are distinct from the pure "Non-Exercise" regions (e.g. solid lines at $t = 6.5k, 11.5k$): the data mean and variance appear to be intermediate values between "Pushup" and "Non-Exercise" pure states, showing the model's ability to identify gradual transitions. Results from other MSR time series are included in the supplement.

We further visualize the learned state vectors for both our model and the baseline over time for the BT data in Fig. 5, revealing the benefits of our approach for transition modeling. This dataset has two pure states (stand, run), and transition periods can be clearly seen where the subject accelerates and decelerates between each running segment (e.g. Fig. 5(a) $t = 750, 1300$). The GMM interpolation model in Fig. 5(b) identifies the alternating discrete states, however all of the transition regions appear identical, switching between states almost instantly. Only our Wasserstein barycentric model in Fig. 5(d) captures the varying rates of acceleration and deceleration in the transition regions. This

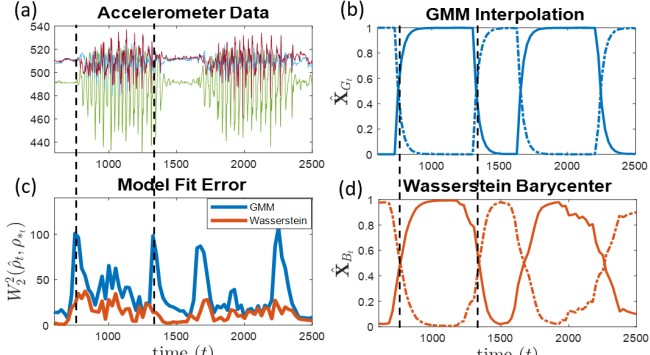

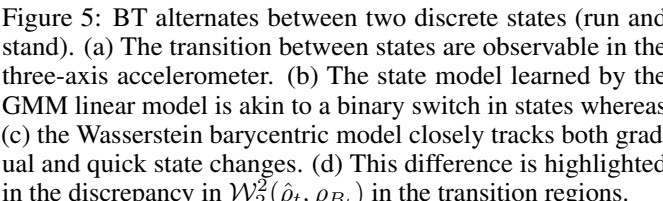

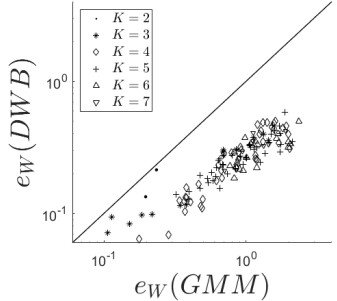

Figure 5: BT alternates between two discrete states (run and stand). (a) The transition between states are observable in the three-axis accelerometer. (b) The state model learned by the GMM linear model is akin to a binary switch in states whereas (c) the Wasserstein barycentric model closely tracks both gradual and quick state changes. (d) This difference is highlighted in the discrepancy in $\mathcal{W}_2^2(\hat{\rho}_t, \rho_{B_t})$ in the transition regions.

Figure 6: Comparison for each MSR time series between the GMM and Wasserstein barycentric interpolation model (DWB) under the $e_W$ evaluation metric. Points below $y = x$ indicate a better data fit for the Wasserstein model.

discrepancy between the models is highlighted in Fig. 5(c), where the improvement of the Wasserstein relative to the GMM model is accentuated in the transition regions.

**Quantitative comparison to GMM linear interpolation.** Fig. 6 shows how our Wasserstein barycentric interpolation model improves data fit quality compared to the GMM linear interpolation model, as shown by the decrease in the $e_W$ metric across all of the 126 MSR time series. Evaluation with respect to $e_{nll}$ (lower is better) shows similar improvement with an average of 1.02 for our Wasserstein barycenter model versus 1.50 for the GMM model. Because the interpolated distributions $\rho_{B_t}$ and $\rho_{G_t}$ are equivalent when the system exists in a pure-state, the benefits of our displacement-interpolation barycentric model come from improved fit during the transition periods.

**Quantitative comparison to Deep State Space (DSS).** As shown in Tab. 2, when given a similar number of parameters, our DWB method outperforms the DSS model regardless of the evaluation metric chosen. When given many more parameters, DSS still has worse $e_W$ but better $e_{nll}$ scores. This difference can be attributed to the fact that the DSS objective is minimized with respect to $e_{nll}$ whereas the DWB objective is minimized with respect to $e_W$. These results suggest that in terms of characterizing a time series' underlying data distribution, our method is competitive with deep learning methods.

|         | DWB $p = 91$ | DSS $p = 2(94)$ | DSS $p \approx 2(88k)$ |
|---------|:---:|:---:|:---:|
| $e_W$   | **0.27** | 4.34 | 3.07 |
| $e_{nll}$ | 1.02 | 1.49 | **-0.508** |

Table 2: Evaluation of DWB and DSS with 94 and 80k parameters on MSR dataset. DWB performs best according to $e_W$, and better than DSS when comparable number of parameters are used under $e_{nll}$.

Our DWB approach has the added benefit of learning the pure-states of the system, something that would require additional post-processing of the latent space for the DSS and other continuous state space models.

**Ablation study of the dynamics prior.** We also consider a variation of our model using a (non-mixture) single Beta distribution prior for $\gamma_t$ with parameters $\boldsymbol{a}[k] = 1.1, \boldsymbol{b}[k] = 3.0, \lambda = 10, s = 2.0$. Because this uses a single Beta for learning both stationary and transition dynamics, the model is more sluggish in adapting to fast changing states as seen in the supplement. Under this configuration, the MSR dataset has an average $e_W = 0.50$ compared to the average $e_W = 0.27$ shown in Fig. 6 for the Beta-mixture learnable prior. A sample plot is included in the supplement.

**Riemannian optimization.** The supplement provides experimental results demonstrating improved optimization speed for estimating $\boldsymbol{\Theta}$ using the Riemannian product manifold discussed in Sec. 5 compared to using the standard Euclidean geometry.

**Hyperparameter sensitivity.** We explore the sensitivity of the results to variations in two key hyperparameters: the scalar $\lambda$ that controls the strength of the data term during learning and the pure state variance $s$, whose inverse also plays the role of a regularization parameter in (9). From Tab. 3, increasing $\lambda$ generally improves the model fit as is expected from (9). For $\lambda \geq 100$, there is an

optimal value $s = 1.0$ for the MSR dataset which corresponds to "reasonable" pure state distributions seen in Fig. 7. For $s$ too small, the distributions are constrained to the centroid of the data. For $s$ too large, the pure state distributions become disjoint from the data themselves, a result allowed by the simplicial structure of the model. In this regime the the barycentric state vector moves away from the vertices towards the interior of the simplex causing more perceived uncertainty between states.

**Sensitivity to initialization.** The supplement includes a plot showing similar results to Fig. 6 obtained when initializing $\Theta$ using ground truth activity labels, suggesting our chosen initialization is unbiased.

## 7   Conclusions and Future Work

Addressing recent trends in technology where the sampling rate of sensors can capture both stationary and transient behaviors of the system, we propose a dynamical Wasserstein barycentric model (DWB) to learn both pure-state emission distributions and the time-varying state vector under a displacement-interpolation transition model between states in an unsupervised setting. For applications such as human activity recognition where transitions are often gradual, the displacement-interpolation given by the Wasserstein barycenter fits data more accurately than the mixture transition model commonly used in the literature. The proposed method can be applied to a wide range of time-series problems including segmentation, clustering, classification, and estimation.

As further contributions, we provide a dynamical state-evolution model of barycentric weight vectors over time. Inspired by previous work in state-space and Bayesian domains, this simplex random walk applies to other temporal modeling applications requiring simplex-valued representations. We also show how tailoring the optimization geometry to the problem leads to improved convergence speeds.

**Limitations.** Due to the need to estimate pure-state distributions in a time-sensitive manner,

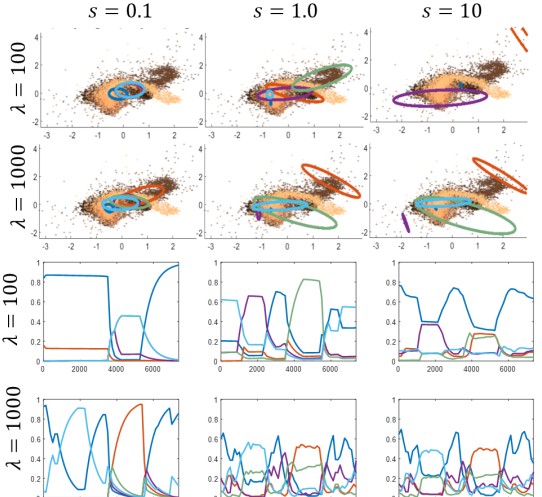

Figure 7: (a) Estimated pure state distributions and (b) estimated state for MSR data varying $\lambda$, $s$

| $\lambda \backslash s$ | 0.01 | 0.1 | 1.0 | 10 | 100 |
|---|---|---|---|---|---|
| 0.1 | 596 | 601 | 581 | 395 | 434 |
| 1.0 | 597 | 593 | 455 | 389 | 378 |
| 10 | 572 | 533 | 284 | 203 | 214 |
| 100 | 521 | 362 | **139** | 157 | 173 |
| 1000 | 410 | 175 | 128 | 137 | 137 |

Table 3: Hyperparameter sensitivity results. Mean $e_W$ for 25 MSR time series varying $\lambda$, $s$. Reported results in this paper use $\lambda = 100, s = 1.0$.

our method is primarily useful for applications where the data is low-dimensional and densely sampled. We have assumed knowledge of the true number of states; in practice at least an upper bound might be known but model size selection remains an open problem. Moreover, we have made a strong assumption that all distributions are Gaussian primarily to leverage the associated geometry and simplify the barycenter computation and optimization.

**Future Work.** Because the Wasserstein barycenter in (3) is defined for any set of distributions with finite second moment (Peyré and Cuturi, 2019), the displacement-interpolation data model outlined in this work can in principle be extended to non-Gaussian distributions. Constructing tractable algorithms based on non-parametric pure state models is certainly an interesting task. We also observe that the simplex random walk is amenable to natural extensions. For example, in our work, we assume that the transition parameters $\gamma_t$ are IID over time. However, by coupling these parameters, we can obtain higher-order smoothness in the simplex-state vector's trajectory.

Finally, the only barrier for making (9) a true likelihood is building a probabilistic model for the model fit based on the Wasserstein distance to the observed data. This has proven difficult to implement as normalization factor in (10) is dependent on the reference distribution. However, with this modification, we can use posterior analysis to properly assess uncertainty in the model and aid in setting the model parameters including the total number of states $K$, which currently is assumed to be known a-priori.

# 8 Acknowledgements

This research was sponsored by the U.S. Army DEVCOM Soldier Center, and was accomplished under Cooperative Agreement Number W911QY-19-2-0003. The views and conclusions contained in this document are those of the authors and should not be interpreted as representing the official policies, either expressed or implied, of the U.S. Army DEVCOM Soldier Center, or the U.S. Government. The U. S. Government is authorized to reproduce and distribute reprints for Government purposes notwithstanding any copyright notation hereon.

We also acknowledge support from the U.S. National Science Foundation under award HDR-1934553 for the Tufts T-TRIPODS Institute. Shuchin Aeron is supported in part by NSF CCF:1553075, NSF RAISE 1931978, NSF ERC planning 1937057, and AFOSR FA9550-18-1-0465. Michael C. Hughes is supported in part by NSF IIS-1908617. Eric L. Miller is supported in part by NSF grants 1934553, 1935555, 1931978, and 1937057.

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
