# OpenReview forum: "Dynamical Wasserstein Barycenters for Time-series Modeling"
_NeurIPS.cc/2021/Conference — NeurIPS 2021 Poster_

### Official Review · Reviewer_nnyz · 2021-07-06

**Rating:** 7
**Confidence:** 4

**Summary:**

This paper deals with the problem of modelling continuous-space time series as sequences of high-level hidden states.
In such models, transitions are usually represented as multinomial mixtures over hidden states.
In this work, another approach is considered in which transition distributions are modelled as Wasserstein barycenters between pure distributions attached to hidden states.
In the case of Gaussian distributions, such barycenters are also Gaussians, and when the match between empirical distributions and those model-predicted distributions is assessed using Wasserstein distance, closed-form formulations can be used.
From those observations, a loss is derived that can be optimized by alternating between (i) gradient descent on the "innovation" parameters $H$ and $\Gamma$ and (ii) gradient descent on a Riemannian manifold.

**Ethics Review Area:**

["Research Integrity Issues (e.g., plagiarism)"]

**Limitations And Societal Impact:**

The authors have included a section about the limitations of their approaches, which is sound.
They have not discussed potential negative societal impact of their work which makes sense since, even if their method can be used for human activity recognition, their contribution does not seem to introduce new societal threats compared to the existing literature in the field.

**Main Review:**

The paper is well motivated, the introduction is insightful and Figure 1 is a very good illustration of the benefits of the method.
The main idea (using Wasserstein barycenters in place of Euclidean ones to better model transitions between states) makes sense and the experiments conducted tend to validate this idea.
Though the paper clearly illustrates the benefits of this approach for human activity recognition, experiments on other kinds of data would have been a plus to better motivate the significance of the contribution outside this specific application setting.

In the proposed approach, one needs to compute the Wasserstein distance between an empirical Gaussian distribution at time $t$ and the Wasserstein barycenter at the same timestamp.
Since only a single sample is observed at time $t$, the parameters of the empirical Gaussian distribution are estimated using a window centered at $t$.
I have two questions related to this:
* How critical is the choice of the window size $n$?
* Why rely on this window centered in $t$ instead of evaluating the likelihood of the observation $y_t$ under the distribution $\rho_{B_t}$?

Below are some minor remarks:
* In Fig 1 (d): the indexes of the Gaussians don't seem to match those of the estimated states
* The Figure provided in Appendix that compares single-component and two-component Beta distributions for $\Gamma_t$ is interesting and would deserve a place in the paper
* It is a pity that Figure 2 does not present prior-related parameters stored in $H$

**EDIT: I have read author rebuttal and it confirms my positive opinion on this paper (and answers my questions related to the use of a sliding window).**

**Time Spent Reviewing:**

5

---

> ### Author Response · Authors · 2021-08-11
> **Author Response**
>
> We thank the reviewer for their time, feedback and for pointing errors and improvements in our original submission, we will correct the figures and discussion as suggested.
>
> **Re “How critical is the choice of the window size?”**
> Since only a single instance of a time series is provided we must rely on adjacent samples to estimate the underlying distribution at a given time. A larger window size ensures some robustness in the estimation of the underlying distribution to outliers, non-stationarity, and even periodicity (as is the case of human activity) in the data. The tradeoff with large window size is reduced temporal accuracy. Since a main motivation of our model is to characterize the distribution during the transition, large windows may blur much of the interesting dynamics that occur in these transitions.
>
> Additionally, since the parameters of the transition dynamics are learnable, when the window size is small there is a real risk where the model converges to a solution where the learned state changes drastically on a per-sample scale instead of learning the higher structure of the changes, a problem in time-series modelling sometimes called fast-switching.
> We set our window size ($n=100$ for BT, $250$ for MSR) with prior knowledge on the human activity data that the activity transitions of interest occur on the order of seconds. Should these parameters be decreased (eg by a factor of 10), we could potentially observe interesting structures of states being broken down into sub-states. To do this, the number of learned states $K$ would have to be increased.
>
> With that said, the reviewer’s suggestion brings up an interesting area of investigation where instead of relying on large $n$ to mitigate fast-switching, we instead fix the Beta parameters to be $a>>b>1$ (strongly biased towards small steps) to control the scale at which transitions occur. This approach may be more applicable in settings where the state moves slowly but at a more or less constant rate rather than what is observed in the human activity case which alternates between periods of stationarity and transition, which motivates the bimodal-beta transition model.
>
> **Re “Why rely on this window centered in t”**
> Since the samples between adjacent sliding windows are largely redundant, in our experiments, we are able to reduce the number of time points that the model is evaluated by skipping $n/4$ points between sliding windows (for a window size $n$). This results in a reduction to the complexity of the model by decimating the number of time points where the state is updated. An approach that evaluates the likelihood of $y_t$ given the $\rho_{B_t}$ would require the estimation of the state at sample, which in the case of the MSR dataset (15,000 samples) was computationally restrictive.
>
> A related approach would be to evaluate according to the likelihood of all the samples in a window assuming that they are all drawn IID $\rho_{B_t}$. To this end, we have repeated our evaluation on the MSR dataset using the log-likelihood of the data in the empirical window assuming the 501 samples are drawn IID from $\rho_{(B,G) t}$. Under this evaluation, the average log-likelihood is -655 for $\rho_{B_t}$ and -748 $\rho_{G_t}$, showing that the Wasserstein barycentric model is a better match to the empirical data compared to the Gaussian model. We will include this evaluation in the final draft.

---

> > ### Comment · Reviewer_nnyz · 2021-08-16
> > **Post-rebittal**
> >
> > I would like to thank the authors for their rebuttal, which confirms my positive opinion on this paper.

---

### Official Review · Reviewer_r2KJ · 2021-07-13

**Rating:** 5
**Confidence:** 3

**Summary:**

To study systems with pure-states and transition periods between pure-states, the authors propose a dynamical model which uses multivariate normal distributions to represent pure-states and characterizes transitions between states via the displacement-interpolation  specified by the Wasserstein barycenter. Learning parameters of pure-state distributions and simplex-valued barycentric weight vector evolving over time or dynamics of the system are performed in a joint way with the help of ADAM and Riemannian line search. Experimental results on human activity datasets are promising.


**Main Review:**

Pros:
1. Employing Wasserstein barycenter for state transition and interpolation sounds a very interesting idea.
2. System dynamics can be characterized by simplex-valued barycentric weight vector evolving over time and learned from data. Such kind of modeling techniques might be applicable to other problems.

Cons
1. The authors employ Beta distribution to model prior over innovations \gamma_t or random factors in a dynamic system. I am wondering whether such kind of prior has enough descriptive power to model complex time series in reality with characteristics like non-stationary, heavy tail distribution, etc. In addition, distributions of pure-states are modeled using Gaussian distribution which has nice analytical properties and close-form solution for the Wasserstein barycenter computation. However, in practice it may not be optimal solution for state space modeling due to its limited discriminative power regarding complicated data distribution.
2. Algo. 1 is a block-wise coordinate descent algorithm in essence. Convergence analysis is  expected which may help readers to understand how challenge the problem is and to what extent solutions found by the algorithm is optimal.
3. Fig. 1 is not intuitive. From it I am not sure why the Wasserstein barycenter is superior to linear interpolation.
4. In the experiments, GMM is employed as a baseline method which is not sufficient. Are there any other state-of-the-art methods in literature which could be used to illustrate the advantages of the proposed method?



**Time Spent Reviewing:**

3

---

> ### Author Response · Authors · 2021-08-11
> **Author Response**
>
> We thank the reviewer for their time and feedback.
>
> **Re (1) “... Beta distribution … descriptive power ...”**
> Our approach draws inspiration from white-noise driven models (e.g. Kalman filters, linear dynamical systems). The use of additive, often white, Gaussian noise to drive these models has proven quite useful for the many problems where the state space is unconstrained.  In this work, we are concerned with the case where the state vector was constrained to lie on the simplex to model the continuous transition between a set of pure states.  Motivated by the ideas in [Nguyen 2020: On a class of random walks in simplexes] and [Ren 2008: The dynamic hierarchical Dirichlet process], the beta "noise" appears to be a useful, if not natural, method for driving the state space model in a manner which simultaneously enforces the simplex constraint.
>
> **Re "...Beta distribution … characteristics like non-stationary,..."**
> The Beta mixture on the transition dynamics is indeed fixed across time thus modeling the transition dynamics as stationary. Making these parameters time-varying does open up interesting applications for non-stationary transition dynamics. However, this assumption of stationary transition dynamics is shared in many baseline discrete and continuous state space models and thus we believe it to be an adequate starting point given the context of this work.
>
> **Re "...heavy-tail distribution…"**
> Unlike continuous state space models where the state belongs to some unconstrained vector space, the state in our method is constrained to the simplex. Since the transition parameters are constrained to [0,1], the need for heavy-tailed distribution does not seem to be as necessary to fully describe all possible state transitions. Other distributions on [0,1] can be explored and compared, however, for our context, we believe the Beta distribution provides adequate flexibility.
>
> **Re "... In addition, … Gaussian distribution…"**
> We are not claiming (nor do we believe) that a Gaussian model for the pure state distributions is optimal in all, perhaps even most, cases. For example, when underlying distributions are multimodal, a Gaussian model would be far from optimal.  To the best of our knowledge however, the general approach we are proposing of a time-varying, simplicial constrained barycentric model for time series analysis has not been considered to date.  To gain initial understanding, we chose in this work to limit consideration to multivariate Gaussian pure state distributions. Even with this assumption, the empirical performance of the Gaussian-based model was quite good at characterizing the actions and transitions between actions in human activity time series analysis.
>
> Reflecting the sentiments of the reviewer, a key focus of our work since submitting the paper under review has been extending the approach to time series where the underlying distributions of pure states and transitions between pure states are non-Gaussian.
>
> **Re (2) “Algo. 1 is a block-wise coordinate…”**
> The optimization algorithm involves alternating the minimization over multiple different manifolds. Thus the analysis is non-trivial. Nonetheless, we will investigate this matter and thank the reviewer for highlighting the importance of proper analysis regarding the optimality of the obtained solution.
>
> **Re (3) “Fig. 1 is not intuitive…”**
> We acknowledge that the top graphic in figure 1 (b) is incorrect in that it should display a mixture of $\rho_1$, $\rho_2$.  We also apologize for any confusion arising from the conflicting labels in the plots of figure 1 (d).
> We do wish to emphasize that figure 1 is intended to highlight the _difference_ between the two interpolation approaches and not to necessarily show _superiority_ of the Wasserstein barycentric model. We appreciate the reviewer’s comments for raising the need to make this differentiation in figure 1 clearer in the final version.
>
> The claim of our paper is that for time series where transitions follow a displacement model, the Wasserstein barycentric model is preferred. This is often the case in applications where one distinct data class morphs continuously to another. This could apply to many physical systems such as protein folding, phase transition, and, as shown in this paper, human activity. In each of these cases, there exists discrete states of the system as well as the smooth continuous flow in the transition between pure-states.
>
> On the other hand, for time series where transitions are effectively “instantaneous”, the GMM approach would be preferred. Such models are also of use in  discrete labeling problems (eg classification of cats and dogs) where there is little sense of introducing a notion of being “in between” to labels (eg cat-dog hybrid). In these cases, label uncertainty, which leads to the linear mixing model, is the more natural interpretation of intermediate states.
>
> **Re (4) “In the experiments…”**
> The motivation and main focus of this work is to jointly identify stationary pure states in the system as well as characterize the transitions between pure states (assuming a displacement interpolation transition framework) given a fixed-length observed time series. We are not aware of any other work that addresses the modeling of time series with this joint goal. In our view, the most similar previous methods in time series modelling either assume instantaneous transitions between discrete states (eg change point detection or time series clustering), or smooth motion in a continuous state space where no discrete states are identified (eg traditional state space models). Therefore, our natural comparison is to address the linear mixture transition model inherent to many baseline switching-state-space methods.
>
> The experiments in the paper were chosen to highlight our model’s ability to more accurately identify the system’s pure states and track the underlying distribution of samples in transition regions, show the efficacy on a complex dataset, and provide some intuition on the model hyperparameters under the application of human activity analysis. However, we agree that comparisons to other works in the literature would be beneficial.
>
> To that end, we are currently working on adding comparisons of our approach with a neural state-space model (Krishnan 2016: https://arxiv.org/pdf/1609.09869.pdf). Where our approach uses a simplex state space to represent transitions between learned pure states, their approach simply uses a continuous state-space model without the added condition of identifying pure states within their state space. They also offer a more flexible data-generating model than ours. Since the output of the deep state-space model differs from our approach we will take careful considerations to ensure fair comparison.

---

> > ### Comment · Reviewer_r2KJ · 2021-08-23
> > **Thank you for the clarification.**
> >
> > The comments on the beta distribution makes sense.

---

> > > ### Author Response · Authors · 2021-08-27
> > > **Additional Benchmarks**
> > >
> > > **Re: other state-of-the-art method**
> > >
> > > Following the feedback from multiple reviewers, we compare our approach to (Krishnan et al. “Structured Inference Networks for Nonlinear State Space Models”, AAAI 2017), which extends Gaussian state space models by modeling the parameters of the state transition and emission distributions (e.g. Gaussian) with deep neural networks. The networks are trained by maximizing the log likelihood via tractable amortized variational lower bound. We will refer to this model as DeepSS, and our dynamical Wasserstein barycentric approach as DWB.
> > >
> > > Both DWB and DeepSS are motivated by time series that contain continuous state transitions, and are effective in characterizing the time-varying underlying data distribution. Thus DeepSS represents a reasonable “state of the art” method for us to compare our model to. However, unlike the DeepSS model, our DWB model is motivated by a class of time series where the system  *gradually transitions* between a *set of discrete pure states*. We address these added considerations in the design of our model by learning a discrete set of pure-states and defining an *easily interpretable latent state*.
> > >
> > > We evaluate using the Microsoft research (MSR) human activity dataset as discussed in our work. Both models are trained and tested on each time series, and evaluated under two metrics. The first, as in eq (11), is the average Wasserstein distance between emission and empirical distributions (W2). The second is the average negative log-likelihood (NLL) of the data. We highlight that DWB is minimized with respect to W2 and DeepSS is minimized with respect to NLL.
> > >
> > > Additionally, the maximum number of parameters for the DWB model for the MSR dataset is $p=88$ ($d=3, K=8,  O(d^2 K)$). Therefore, we evaluate the results of the DeepSS model under two different settings: one where the transition and emission neural network has 94 parameters *each* and one using the default settings given by https://github.com/clinicalml/dmm where $p \approx 80,000$. The exact model details are given later. Due to time constraints reported results are averaged over 3 arbitrarily selected time series from the larger MSR dataset.
> > >
> > > |      | DWB ($p=88$) | DeepSS ($p=94*2$) | DeepSS ($p \approx  80k*2$) |
> > > | --- | --- | --- | --- |
> > > | W2   |  **0.333** | 4.340 | 3.076 |
> > > | NLL  |  1.022 |	 1.488 |  **-0.508** |
> > >
> > > **As shown in the above table, when given a similar number of parameters, our DWB method outperforms the DeepSS model regardless of the metric chosen**. However, when given 1000x more parameters, DeepSS still has worse W2 but better NLL scores. This can be attributed to the fact that the DeepSS model is minimized with respect to NLL and the DWB is minimized with respect to W2. While a complete evaluation of the whole dataset is warranted, these preliminary results show that in terms of characterizing the underlying distribution of the time series, our method is comparable to state of the art deep learning methods.
> > >
> > > We emphasize that in addition to the characterization of the underlying distribution shown in the above evaluation, our DWB approach *identifies a set of discrete pure states in the system and yields a directly-interpretable per-sample state representation*; the Wasserstein barycentric mixing weights for each pure-state is directly given by the corresponding component of the simplex-state vector. In contrast, there is no direct interpretation of the latent state and no equivalent comparison to the learned pure-state in the DeepSS model. Additional ad hoc processing (e.g. clustering) of the DeepSS latent state space would be required to achieve this, which is beyond the scope of our present paper.
> > >
> > > The following table highlights additional similarities and differences between the two models
> > >
> > > |	| DWB | DeepSS |
> > > | --- | --- | --- |
> > > | State Space | $K$-simplex | $R^n$ |
> > > | State Transition Dynamics | Beta mixture | Gaussian distribution |
> > > | State Transition Parameters | Learnable Beta parameters | neural network parameterizing mean and diagonal covariance of Gaussian |
> > > | Emission distribution model | Gaussian with full covariance | Gaussian with diagonal covariance |
> > > | Number of learned parameters | $O(d^2 K)$ $d$=data dimension, $K$=# clusters | $O(m^2)$ $m$=NN hidden layer size |
> > >
> > > As mentioned, we run the DeepSS with two parameter settings. The first uses 2 hidden layers each with 5 neurons for a total of 94 learned parameters for *each* transmission and emission networks. The second uses 3 hidden layers for the transition and emission networks with a hidden state dimension of 200 for a total of $p=82,607$ for *each* neural network. In both cases, the latent space has dimension (K-1) (to match the dimensionality of the K-simplex), an RNN of 2 layers and 600 nodes each is used as the variational approximation network, and training occurs over 1000 epochs with a learning rate of 0.008. Plots of the variational lower bound show convergence under these conditions

---

> > > > ### Author Response · Authors · 2021-09-02
> > > > **Comparison Update**
> > > >
> > > > The table below provides the updated comparison between DWB and DeepSS models when accounting for the full MSR dataset (n=126).  There is no major change in the discussion provided above given this expanded experiment. These results will be incorporated into the final version of our submission.
> > > >
> > > > |      | DWB ($p=88$) | DeepSS ($p=94*2$) | DeepSS ($p \approx  80k*2$) |
> > > > | --- | --- | --- | --- |
> > > > | W2   |  **0.313** | 3.323 | 4.034 |
> > > > | NLL  |  1.307 |	 2.36 |  **-0.714** |

---

### Official Review · Reviewer_Vw6y · 2021-07-16

**Rating:** 5
**Confidence:** 4

**Summary:**

The authors present a model for the state of a system over time as a Wasserstein barycenter as a mixture between a (known) number of pure states modeled as Gaussian distributions. Modeling states as Gaussian distributions results in more efficient computation given the closed form of Wasserstein barycenters between Gaussian distributions.

**Limitations And Societal Impact:**

Yes

**Main Review:**

This model is empirically validated by comparing to a GMM model over two time series datasets with state annotations using the a Wasserstein distance error between predict and ground truth distributions.

Question: Why should the Beta distribution be unimodal? I would think that there are many instances when it should not be unimodal, and could more often have a number of modes equal to the number of pure states.

This is an interesting idea to improve over gaussian mixture models in some time series. However, this manuscript does not relate to any of the work on time series in the ML conference cycle and is not compared to any recent methods, and further is only applied to a very specific time series. It would be interesting to see how this model performs against some of the more recent deep learning models and on a wider variety of datasets. Without this it is difficult to evaluate where this model fits into the broader literature.

N-BEATS: https://arxiv.org/abs/1905.10437
M4 Competition:https://www.sciencedirect.com/science/article/pii/S0169207019301128
-----------------
Post author response update:
Thank you for clearing up these issues and the problem setting you are tackling in this paper. I now understand that this is different from the traditional time series interpolation and extrapolation problem. Since a large number of the results use error from known states as ground truth this seems quite similar to me. I agree that it would be a good idea to compare to deep state-space models as mentioned by the authors and more recent work in this line.

I am updating my score to weak reject as I still believe it would be useful to have comparison to other related work on deep state space modeling and further comparison to DeepSS (although preliminary results provided by the authors looks promising).

**Time Spent Reviewing:**

3

---

> ### Author Response · Authors · 2021-08-11
> **Author Response**
>
> We thank the reviewer for their time and feedback.
>
> **Re "Why should the Beta distribution be unimodal?..."**
> We thank the reviewer for providing the opportunity to clarify our use of beta random variables within the context of our simplex-constrained state space model.  Toward this end, consider Eq. (8) in the special case where all of the $w[k]$ are identically zero.  In this case, we would be using IID beta random variables to model the incremental motion of the state vector to each of the “pure state” vertices in the simplex.  Roughly speaking each beta is the fraction of the distance moved from the current state to each vertex. As such, we have found it most useful to keep these random variables unimodal.  Shifting the mode toward zero means small steps from the current state toward any given vertex.  A mode closer to unity would provide for larger steps.  While it is possible to create bimodal (but not tri- or high modal) betas, the resulting PDFs diverge at zero and/or one and may lead to undesired behaviors such as all-or-nothing transition dynamics.
>
> The use of the two component mixture in Eq. (8) where the $w[k]$ are not zero provides additional flexibility by allowing for both “small” as well as “large” motions from the current state to any vertex.  Because there are K such bimodal random variables used for each time, we are in effect using a multimodal beta model similar to the one suggested by the reviewer.  In fact, the number of modes is twice the number of vertices to account for “fast” and “slow” dynamics for each vertex.
>
> **Re "However, this manuscript does not relate…"**
> The motivation and main focus of this work is to _identify pure states_ and _characterize the gradual transitions_ between them given a fixed-length observed time series. As we are not concerned with forecasting future values of our time series, we do not think the suggested M4 datasets or N-BEATS baseline is the most suitable way to evaluate our work.
>
> We are not aware of any other work that addresses the modeling of time series with this joint approach for characterizing stationary and transition periods nor are we aware of any works that  use this _gradual transition_ perspective. In our view, the most similar previous methods in time series modelling either assume instantaneous transitions between discrete states (e.g. change point detection or time series clustering), or smooth motion in a continuous state space where no discrete states are identified (e.g. state space models). Therefore, our natural comparison is to address the linear mixture transition model inherent to many baseline switching-state-space methods.
>
> Nonetheless, we are currently working on adding comparisons of our approach with a neural state-space model (Krishnan 2016: https://arxiv.org/pdf/1609.09869.pdf). Where our approach uses a simplex state space to represent transitions between learned pure states, their approach simply uses a continuous state-space model without the added condition of identifying pure states within their state space. They also offer a more flexible data-generating model than ours. Since the output of the deep state-space model differs from our approach we will take careful considerations to ensure fair comparison.

---

> > ### Author Response · Authors · 2021-08-27
> > **Additional Benchmarks**
> >
> > **Re: compared to recent methods**
> >
> > Following the feedback from multiple reviewers, we compare our approach to (Krishnan et al. “Structured Inference Networks for Nonlinear State Space Models”, AAAI 2017), which extends Gaussian state space models by modeling the parameters of the state transition and emission distributions (e.g. Gaussian) with deep neural networks. The networks are trained by maximizing the log likelihood via tractable amortized variational lower bound. We will refer to this model as DeepSS, and our dynamical Wasserstein barycentric approach as DWB.
> >
> > Both DWB and DeepSS are motivated by time series that contain continuous state transitions, and are effective in characterizing the time-varying underlying data distribution. Thus DeepSS represents a reasonable “state of the art” method for us to compare our model to. However, unlike the DeepSS model, our DWB model is motivated by a class of time series where the system  *gradually transitions* between a *set of discrete pure states*. We address these added considerations in the design of our model by learning a discrete set of pure-states and defining an *easily interpretable latent state*.
> >
> > We evaluate using the Microsoft research (MSR) human activity dataset as discussed in our work. Both models are trained and tested on each time series, and evaluated under two metrics. The first, as in eq (11), is the average Wasserstein distance between emission and empirical distributions (W2). The second is the average negative log-likelihood (NLL) of the data. We highlight that DWB is minimized with respect to W2 and DeepSS is minimized with respect to NLL.
> >
> > Additionally, the maximum number of parameters for the DWB model for the MSR dataset is $p=88$ ($d=3, K=8,  O(d^2 K)$). Therefore, we evaluate the results of the DeepSS model under two different settings: one where the transition and emission neural network has 94 parameters *each* and one using the default settings given by https://github.com/clinicalml/dmm where $p \approx 80,000$. The exact model details are given later. Due to time constraints reported results are averaged over 3 arbitrarily selected time series from the larger MSR dataset.
> >
> > |      | DWB ($p=88$) | DeepSS ($p=94*2$) | DeepSS ($p \approx  80k*2$) |
> > | --- | --- | --- | --- |
> > | W2   |  **0.333** | 4.340 | 3.076 |
> > | NLL  |  1.022 |	 1.488 |  **-0.508** |
> >
> > **As shown in the above table, when given a similar number of parameters, our DWB method outperforms the DeepSS model regardless of the metric chosen**. However, when given 1000x more parameters, DeepSS still has worse W2 but better NLL scores. This can be attributed to the fact that the DeepSS model is minimized with respect to NLL and the DWB is minimized with respect to W2. While a complete evaluation of the whole dataset is warranted, these preliminary results show that in terms of characterizing the underlying distribution of the time series, our method is comparable to state of the art deep learning methods.
> >
> > We emphasize that in addition to the characterization of the underlying distribution shown in the above evaluation, our DWB approach *identifies a set of discrete pure states in the system and yields a directly-interpretable per-sample state representation*; the Wasserstein barycentric mixing weights for each pure-state is directly given by the corresponding component of the simplex-state vector. In contrast, there is no direct interpretation of the latent state and no equivalent comparison to the learned pure-state in the DeepSS model. Additional ad hoc processing (e.g. clustering) of the DeepSS latent state space would be required to achieve this, which is beyond the scope of our present paper.
> >
> > The following table highlights additional similarities and differences between the two models
> >
> > |	| DWB | DeepSS |
> > | --- | --- | --- |
> > | State Space | $K$-simplex | $R^n$ |
> > | State Transition Dynamics | Beta mixture | Gaussian distribution |
> > | State Transition Parameters | Learnable Beta parameters | neural network parameterizing mean and diagonal covariance of Gaussian |
> > | Emission distribution model | Gaussian with full covariance | Gaussian with diagonal covariance |
> > | Number of learned parameters | $O(d^2 K)$ $d$=data dimension, $K$=# clusters | $O(m^2)$ $m$=NN hidden layer size |
> >
> > As mentioned, we run the DeepSS with two parameter settings. The first uses 2 hidden layers each with 5 neurons for a total of 94 learned parameters for *each* transmission and emission networks. The second uses 3 hidden layers for the transition and emission networks with a hidden state dimension of 200 for a total of $p=82,607$ for *each* neural network. In both cases, the latent space has dimension (K-1) (to match the dimensionality of the K-simplex), an RNN of 2 layers and 600 nodes each is used as the variational approximation network, and training occurs over 1000 epochs with a learning rate of 0.008. Plots of the variational lower bound show convergence under these conditions

---

> > > ### Author Response · Authors · 2021-09-02
> > > **Comparison Update**
> > >
> > > The table below provides the updated comparison between DWB and DeepSS models when accounting for the full MSR dataset (n=126).  There is no major change in the discussion provided above given this expanded experiment. These results will be incorporated into the final version of our submission.
> > >
> > > |      | DWB ($p=88$) | DeepSS ($p=94*2$) | DeepSS ($p \approx  80k*2$) |
> > > | --- | --- | --- | --- |
> > > | W2   |  **0.313** | 3.323 | 4.034 |
> > > | NLL  |  1.307 |	 2.36 |  **-0.714** |

---

### Official Review · Reviewer_p5FH · 2021-07-27

**Rating:** 6
**Confidence:** 4

**Summary:**


Summary
=======

Context:
--------
This paper proposes a reconstruction model for time series that are modeled using:
- a sequence of K distributions (\rho_1, ..., \rho_K) each representing a certain "state"
- a transition vector of weights (x_t)_t \in Simplex_K that weight the different states at each
time point.

Starting from observed vector valued data y_1, ..., y_T, the goal is to identify the state distributions \rho_k
and the transition weights (x_t).

Contribution
------------
This paper proposes to do so by considering Gaussian distributions (\rho) and modeling each observation
y_t as a sample of the Wasserstein barycenter of (y_1, ..., y_K) weighted by (x_t[0], ... x_t[K]).

The transition states x_t are modeled using a Markov chain with a Beta distribution prior on its parameters (Gamma).
All parameters (Gamma, means and covariances of \rho) are estimated via fitting the data (with the Wasserstein barycenter
loss) and the prior on the parameters.

The proposed model is compared to a GMM (i.e replacing OT with a linear interpolation) on two experiments illustrating
the flexibility of using OT in estimating the transient state weight vector x.


**Main Review:**

review
======

General assessment and main concerns
-----------------------------------

Overall, the paper is well presented and very easy to read. The illustrated examples figs 1-3 play helped a lot.
The proposed idea is simple and intuitive: if samples between states are more likely to smoothly transition from one distribution
to another than simply "disappear / appear" then an OT interpolation makes more sense than a linear one. While the proposed model
as well as its implementation are convincing, my main concern is that the experiments as well as the introduction is a bit too
restricted / niche.

(A) I would have like to know for instance the nature of the data for which such dynamic models apply ? why would
a Gaussian assumption be realistic ? Are there any scenarios where a linear interpolation would be more suited than OT ?

(B) All experiments used do not have ground truth. While this is sometimes inevitable, the performance of a new model should always be assessed
in controlled experiments. The authors resort to using a data fitting term to compare the OT interpolation against the linear one which is biased
since it is literally minimized in their proposed loss function. While the presented evidence is interesting and encouraging, it is inconclusive
at best. These concerns as well as (A) could have been adressed using simulations where the strengths / limits of the proposed model could be
pinpointed.

Specific comments
-----------------

1) Equations (6) and (9) display the W_2 distance. However I believe it should be the Bures metric instead ? Moreover, the empirical measures
\rho_t should be replaced by their empirical moments since the model doesn't exactly fit the empirical observations but their moments instead.
This is even more confusing since it is the same as the fit error (11) which a priori should be using the exact empirical data.

2) The Gaussian approximation is a very strong assumption (perhaps necessary if d and n are large) and OT decomposes as a Euclidean distance between
the means and Bures between the covariances. It would have been interesting to perform comparisons where the covariance term is replaced by another
metric over the cone of PD matrices or even entirely removed. Perhaps discriminating using the means would be enough ?

3) Using (11) as an evaluation metric introduces a systemic bias here since it is quite literally minimized in the proposed method:
it would naturally be lower than any other benchmark.

4) For initialization, why not use the output of the GMM fit entirely and use one mean m_0  and one sigma_0 ?

5) If you use such an initialization where you have initial estimates of all the means and covariances, another benchmark
would be to estimate the weights x_k directly by projecting on the simplex.

6) Were the specified values in L184-L185 manually data driven ?

Minor remarks and typos:
---------------------

- L44 unfinished sentence
- L145 something strange here in this sentence

**Time Spent Reviewing:**

4

---

> ### Author Response · Authors · 2021-08-11
> **Author Response**
>
> We thank the reviewer for their time and feedback.
>
> **Re: (A) “why would a Gaussian assumption be realistic ?”**
> We thank the reviewer for this important point. We are not claiming (nor do we believe) that a Gaussian model for the pure state distributions is suitable in all, perhaps even most, cases. For example, when underlying distributions are multimodal, a Gaussian model would be far from optimal.  To the best of our knowledge however, the general approach we are proposing of a time-varying, simplicial constrained barycentric model for time series analysis has not been considered to date.  To gain initial understanding, we chose in this work to limit consideration to multivariate Gaussian pure state distributions. This choice allowed us to exploit the substantial benefits arising from the many differential geometric closed-form (or close to closed-form) results associated with Gaussian densities. Even with this assumption, the empirical performance of the Gaussian-based model was quite good at characterizing the actions and transitions between actions in human activity time series analysis.
>
> Reflecting the sentiments of the reviewer, a key focus of our work since submitting the paper under review has been extending the approach to time series where the underlying distributions of pure states and transitions between pure states are non-Gaussian.
>
> **Re “[for what data do] dynamic models apply?”**
> “Are there any scenarios where a linear interpolation would be more suited than OT” **
> The model in this paper is motivated by applications where the system under consideration undergoes gradual transitions among a number of what we are calling “pure states.”  In such cases, we have tried to differentiate the displacement interpolation view on transitions from the to the commonly assumed mixture interpolation given by linear (really, convex) combination of distributions.
>
> To recall, the displacement model moves the distribution along the optimal map $T$ joining the two distributions
>
> $$
> \rho_w = (w Id + (1-w)T)_{\sharp}  \rho_1
> $$
>
> Where $\sharp$ denotes the pushforward of a measure, $\rho_2 = T_{\sharp}\rho_1$ and $Id$ is the identity map.
>
> Alternatively, the linear mixture model characterizes the transition between two states as:
>
> $$
> \rho_w = w \rho_1 + (1-w)\rho_2
> $$
>
> For some concrete examples of this distinction: the linear mixing model is often suited in situations when the underlying state space is truly discrete (eg labels of cats, dogs, etc.). Here, if $\rho_1$, $\rho_2$ represent the distribution under cat and dog label respectively, there is little sense of introducing a notion of being “in between” two labels (eg cat-dog hybrid) according to the displacement model. More natural is the interpretation of state given by label uncertainty which yields the distribution provided by linear mixing (eg 50% likelihood of cat, 50% of dog would have a distribution $0.5\rho_1+0.5 \rho_2$). Even in applications where the labels can be embedded in a continuous domain, there are situations where the linear interpolation model is preferable because underlying state transitions follow linear mixing. For example, in word embedding topic models, hybrid topics would likely contain a mixture of the word embeddings associated with each topic.
>
> On the other hand, the displacement interpolation model generally matches applications where one distinct data class morphs continuously to another. This could apply to many physical systems such as protein folding, phase transition, and, as shown in this paper, human activity. In each of these cases, there exists discrete states of the system as well as smooth continuous flow in the transition between pure-states. One consideration is that the sampling rate has to be high enough to observe these intermediary states. For example, while the onset of a disease may be a continuous transition, the transition may not be observable based on the system sampling rate (eg medical checkups).  In the former case, the displacement model would likely be best.  In the latter case, one could argue that a linear mixture model would be preferable.
>
> We emphasize that the difference in these models appears in the nature of the transitions. When the system resides fully in a pure state (state vector is 1-hot), the resulting distribution is identical between both mixture and displacement models. This is highlighted in the beep-test data shown in figure 5. **Re "Perhaps discriminating using the means..."** This is an example where discriminating between just the means is insufficient as the data for two states of standing and running have rather similar mean values but widely differing variances. In this case, both mixture and displacement models accurately characterize the data in the stationary regions. However, during the transitions, the linear model correctly identifies the switching states, but does so in a way that is akin to binary switching (with some smoothness due to constraints on state dynamics). On the other hand, the Wasserstein barycentric model does a better job of resolving the gradual transition as seen by the much improved data fit term in figure 5.
>
> **(B)  ...addressed using simulations…”**
> We thank the reviewer for raising this important aspect of using simulated data to better understand the strengths and weaknesses of the approach we are proposing.  Toward that end, we have considered numerical experiments (similar to that included in the supplement regarding optimization simulations) consisting of two randomly generated Gaussian pure-states of varying dimension and a state vector that transitions linearly with constant speed between the two using the displacement interpolation transition model.
>
> In this simple simulated setting, we observe that for all dimensions ($d=2,10,50$), the Wasserstein barycentric model ($\rho_B$) fits the empirical data much better than linear mixing ($\rho_G$), as seen in the table below. The metric in the table is given by Eq. (11).
>
> |    | $\rho_B$ | $\rho_G$ |
> | ---| ------------ | ------------ |
> | d=2  |  0.148 | 0.298 |
> | d=10 | 0.22 | 0.352 |
> | d=50 |  0.656 | 0.795 |
>
> Here we provide a detailed explanation of how the simulated data was generated. The state linear interpolates between the two states over $T=100$ timesteps ($x[0]=linspace(0,1,T), x[1]=linspace(1,0,T)$). The first Gaussian ($\rho_1 = N(m_1, S_1)$) was generated according to $m_1 ~ N(0, I)$, and $S_1$ according to (Davies 2000: Numerically Stable Generation of Correlation Matrices and Their Factors). The second Gaussian was generated by traveling a set distance along a random tangent direction (using Wasserstein geometry). Thus, $E^2(m_1, m_2)=1$ and $B^2(S_1,S_2)=2$, (where E and B are the Euclidean and Bures metric respectively) such that $W^2_2 (\rho_1, \rho_2) = 3$. This was repeated 10 times for each dimension $d=2,10,50$ and reported results are the average. For each time step, an empirical distribution was generated according to $20d$ samples drawn according to the Wasserstein barycentric model. This experiment was performed using a single component beta transition model since the time series consists of a constant speed transition between two states with Beta parameters $a=1.1, b=10$.

---

> > ### Author Response · Authors · 2021-08-11
> > **Author Response (cont)**
> >
> > **Re (1) “Equations (6) and (9) display…”**
> > Because we are considering Gaussian models with non-zero mean vectors, the Wasserstein distance between Gaussians is the sum of the Euclidean distance between the means and the Bures distance of covariances resulting in Eq. (6), and Eq. (9) as stated. The distributions $\rho_t = N(m_t, S_t)$ are in fact characterized by their moments as discussed in section 3. This is used for both the minimization problem in Eq. (6) and Eq. (9) as well as the evaluation in Eq. (11). The reviewer’s suggestion to use the exact empirical data is interesting where one alternative evaluation metric would be the likelihood of the empirical data in a window and is addressed below in Re (3).
> >
> > **Re (2) "...another metric over the cone of PD …"**:
> > As suggested by the reviewer, replacing the Bures metric in Eq. (2) with any metric on the positive definite (PD) cone with tractable and differential computation of the metric and associated barycenter would be valid in our stated model. Two alternatives to the Bures metric are the Frobenius norm, and the Hellinger-based metric proposed in (Bhatia 2020: Matrix Versions of the Hellinger Distance). We ran the same simulated experiment described in (B) substituting these PD metrics for Bures distance in the Gaussian distance computation in Eq. (2) and barycenter computation in Eq. (4). These are denoted in the table below as $\rho_{F}$ and $\rho_{H}$ for the Frobenius and Hellinger distance respectively.
> >
> > |  | $\rho_B$ |  $\rho_F$ | $\rho_H$ |
> > | ---| --- | --- | --- |
> > | d=2  | 0.148 |  0.172 | 0.158 |
> > | d=10 | 0.220  |  0.220  | 0.222 |
> > | d=50 | 0.656 |  0.653 | 0.653 |
> >
> > From this experiment we did not find any significant difference in performance among the three methods across all dimensions. Understanding why this is the case warrants further investigation. In the end, the Bures metric was chosen because of the direct ties to the displacement interpolation model through the Wasserstein barycenter for Gaussian distributions.
> >
> > **Re (3) "Using (11) as an evaluation metric…"**
> > We thank the reviewer for raising the concern of a fair evaluation metric. The systematic bias mentioned is shared between the compared approaches. In both cases we are minimizing with respect to the Wasserstein distance between $\rho_{B_t}$ or $\rho_{G_t}$ for the Wasserstein barycenter or GMM model respectively. Nonetheless, it is a worthy comparison to validate if our conclusions hold when alternative evaluations metrics are used.
> >
> > To that end we report the results averaging across all MSR datasets evaluating using the Wasserstein loss in Eq. (11), the KL divergence, and log likelihood of observations in the empirical window. Evaluating according to the model error according to Eq. (11) $\frac{1}{T} \sum_{t=1:T} W_2^2 ( \rho_t, \rho_{(B,G)t})$ yields 0.27 using the Wasserstein barycentric model and 0.86 for the linear model (smaller is better). Similarly, when using a KL evaluation model $\frac{1}{T} \sum_{t=1:T} KL(\rho_t || \rho_{(B,G)t})$, we get 0.79 for $\rho_{B_t}$ and 14.3 $\rho_{G_t}$ (smaller is better). For the log-likelihood computation we assume the 501 in the empirical window samples are drawn IID from $\rho_{(B,G)t}$. Under this evaluation, the average log-likelihood is -655 for  $\rho_{B_t}$ and -748 $\rho_{G_t}$ (larger is better). Therefore, regardless of the evaluation metric chosen, the Wasserstein barycentric model shows a closer match to the data compared to the linear mixture approach. We will include these results in the final submission and thank the reviewer for their feedback on this matter.
> >
> > **Re (4) "For initialization, why not use…"**
> > To be clear, in our model, $m_0$ and $\sigma_0$ are not connected with the manner in which we initialize the model parameters in the optimization problem.  Rather these quantities are used to specify the prior model over the space Gaussian parameters (mean vectors and covariance matrices) for the pure-states Gaussian. Setting $m_0$ and $sigma_0$ according to L260-261, puts the mean of this prior at a “central” point of the data while maintaining probabilistic properties described in Sec 4.2. In our model, this prior acts as an anchor to ensure that the pure states do not deviate too much from the data, as seen in the ablation study in figure 7 when the variance on this prior is increased. Choosing one component of a GMM would place the mean of the prior on a subset of the data.  We believe that incorporating such a bias into the prior would not be as effective as a central regularization point.
> >
> > To initialize the parameters of the pure state distributions, we make use of prior work in change point detection and time series clustering [Cheng 2020: https://ieeexplore.ieee.org/document/9053344] to identify regions of common activity within a time series. The idea of initializing using a GMM is certainly interesting and one we will explore in the future.
> >
> > **Re (5) "If you use such an initialization…"**
> > We agree that this would be a valid benchmark especially for evaluating the proposed state dynamical model. Due to time constraints, we have decided to focus on implementing an alternate benchmark using a deep neural state space model  (Krishnan 2016: https://arxiv.org/pdf/1609.09869.pdf) during this review period.
> >
> > **Re (6) "Were the specified values..."**
> > The intention was to fix one component of the beta mixture to be unimodal with mean close to 0. The Beta random variable dictates how fast we transition to a particular pure state. Therefore, values near 0 would result in little change in state, which would be expected during stationary periods of the time series. These parameters were just chosen and not optimized for, but reported for reproducibility.

---

> > > ### Comment · Reviewer_p5FH · 2021-08-18
> > > **Post rebuttal**
> > >
> > > I would like to thank the authors for their detailed response.
> > >
> > >  All my concerns were properly addressed. The additional comparisons are certainly worthy of being included in the final version of the paper. I particularly appreciate the discussion in response to (A), i believe this will clarify several points raised by the other reviewers concerning the motivation and usefulness the proposed method and invite the authors to update their introduction accordingly. My only main remaining concern is the lack of benchmarks. One cannot help but wonder how a naive neural net would perform in this task.

---

> > > > ### Author Response · Authors · 2021-08-27
> > > > **Additional Benchmarks**
> > > >
> > > > We thank the reviewer for their consideration and are working to incorporate their feedback to help clarify the presentation.
> > > >
> > > > **Re: Additional benchmarks**
> > > >
> > > > Following the feedback from multiple reviewers, we compare our approach to (Krishnan et al. “Structured Inference Networks for Nonlinear State Space Models”, AAAI 2017), which extends Gaussian state space models by modeling the parameters of the state transition and emission distributions (e.g. Gaussian) with deep neural networks. The networks are trained by maximizing the log likelihood via tractable amortized variational lower bound. We will refer to this model as DeepSS, and our dynamical Wasserstein barycentric approach as DWB.
> > > >
> > > > Both DWB and DeepSS are motivated by time series that contain continuous state transitions, and are effective in characterizing the time-varying underlying data distribution. Thus DeepSS represents a reasonable “state of the art” method for us to compare our model to. However, unlike the DeepSS model, our DWB model is motivated by a class of time series where the system  *gradually transitions* between a *set of discrete pure states*. We address these added considerations in the design of our model by learning a discrete set of pure-states and defining an *easily interpretable latent state*.
> > > >
> > > > We evaluate using the Microsoft research (MSR) human activity dataset as discussed in our work. Both models are trained and tested on each time series, and evaluated under two metrics. The first, as in eq (11), is the average Wasserstein distance between emission and empirical distributions (W2). The second is the average negative log-likelihood (NLL) of the data. We highlight that DWB is minimized with respect to W2 and DeepSS is minimized with respect to NLL.
> > > >
> > > > Additionally, the maximum number of parameters for the DWB model for the MSR dataset is $p=88$ ($d=3, K=8,  O(d^2 K)$). Therefore, we evaluate the results of the DeepSS model under two different settings: one where the transition and emission neural network has 94 parameters *each* and one using the default settings given by https://github.com/clinicalml/dmm where $p \approx 80,000$. The exact model details are given later. Due to time constraints reported results are averaged over 3 arbitrarily selected time series from the larger MSR dataset.
> > > >
> > > > |      | DWB ($p=88$) | DeepSS ($p=94*2$) | DeepSS ($p \approx  80k*2$) |
> > > > | --- | --- | --- | --- |
> > > > | W2   |  **0.333** | 4.340 | 3.076 |
> > > > | NLL  |  1.022 |	 1.488 |  **-0.508** |
> > > >
> > > > **As shown in the above table, when given a similar number of parameters, our DWB method outperforms the DeepSS model regardless of the metric chosen**. However, when given 1000x more parameters, DeepSS still has worse W2 but better NLL scores. This can be attributed to the fact that the DeepSS model is minimized with respect to NLL and the DWB is minimized with respect to W2. While a complete evaluation of the whole dataset is warranted, these preliminary results show that in terms of characterizing the underlying distribution of the time series, our method is comparable to state of the art deep learning methods.
> > > >
> > > > We emphasize that in addition to the characterization of the underlying distribution shown in the above evaluation, our DWB approach *identifies a set of discrete pure states in the system and yields a directly-interpretable per-sample state representation*; the Wasserstein barycentric mixing weights for each pure-state is directly given by the corresponding component of the simplex-state vector. In contrast, there is no direct interpretation of the latent state and no equivalent comparison to the learned pure-state in the DeepSS model. Additional ad hoc processing (e.g. clustering) of the DeepSS latent state space would be required to achieve this, which is beyond the scope of our present paper.
> > > >
> > > > The following table highlights additional similarities and differences between the two models
> > > >
> > > > |	| DWB | DeepSS |
> > > > | --- | --- | --- |
> > > > | State Space | $K$-simplex | $R^n$ |
> > > > | State Transition Dynamics | Beta mixture | Gaussian distribution |
> > > > | State Transition Parameters | Learnable Beta parameters | neural network parameterizing mean and diagonal covariance of Gaussian |
> > > > | Emission distribution model | Gaussian with full covariance | Gaussian with diagonal covariance |
> > > > | Number of learned parameters | $O(d^2 K)$ $d$=data dimension, $K$=# clusters | $O(m^2)$ $m$=NN hidden layer size |
> > > >
> > > > As mentioned, we run the DeepSS with two parameter settings. The first uses 2 hidden layers each with 5 neurons for a total of 94 learned parameters for *each* transmission and emission networks. The second uses 3 hidden layers for the transition and emission networks with a hidden state dimension of 200 for a total of $p=82,607$ for *each* neural network. In both cases, the latent space has dimension (K-1) (to match the dimensionality of the K-simplex), an RNN of 2 layers and 600 nodes each is used as the variational approximation network, and training occurs over 1000 epochs with a learning rate of 0.008. Plots of the variational lower bound show convergence under these conditions

---

> > > > > ### Author Response · Authors · 2021-09-02
> > > > > **Comparison Update**
> > > > >
> > > > > The table below provides the updated comparison between DWB and DeepSS models when accounting for the full MSR dataset (n=126).  There is no major change in the discussion provided above given this expanded experiment. These results will be incorporated into the final version of our submission.
> > > > >
> > > > > |      | DWB ($p=88$) | DeepSS ($p=94*2$) | DeepSS ($p \approx  80k*2$) |
> > > > > | --- | --- | --- | --- |
> > > > > | W2   |  **0.313** | 3.323 | 4.034 |
> > > > > | NLL  |  1.307 |	 2.36 |  **-0.714** |

---

### Decision · Program_Chairs · 2021-09-27

**Decision:**

Accept (Poster)

**Comment:**

The paper proposes dynamical Wasserstein barycenter for modeling time series, observation are considered to be sampled from Bures Wasserstein barycenter of states with time dependent mixing coefficients.  The paper compares wasserstein interpolation of states to linear interpolation of states.

Reviewers were positive about the paper in the sense it demonstrated an advantage on a GMM model with linear interpolation. Several question were raised on why restricting to Bures barycenter and to go beyond gaussianity assumption and on using maybe of other baselines such as neural networks and in providing more experiments on more challenging benchmarks.

I think the paper has a value being a new application of Wasserstein barycenter to time series modeling , and the authors were transparent about the limitations of the work. Weak accept